# Development and Validation of a Food Frequency Questionnaire for Evaluating the Nutritional Status of Patients with Serious Mental Illnesses (DIETQ-SMI) in Bahrain

**DOI:** 10.3390/brainsci14040312

**Published:** 2024-03-26

**Authors:** Haitham Jahrami, Zahra Saif, Achraf Ammar, Waqar Husain, Khaled Trabelsi, Hadeel Ghazzawi, Seithikurippu R. Pandi-Perumal, Mary V. Seeman

**Affiliations:** 1Psychiatric Hospital, Governmental Hospitals, Manama P.O. Box 12, Bahrain; 2Department of Psychiatry, College of Medicine and Medical Sciences, Arabian Gulf University, Manama P.O. Box 26671, Bahrain; zahra-saif@outlook.com; 3Department of Training and Movement Science, Institute of Sport Science, Johannes Gutenberg-University Mainz, 55099 Mainz, Germany; acammar@uni-mainz.de; 4Research Laboratory, Molecular Bases of Human Pathology, LR19ES13, Faculty of Medicine of Sfax, University of Sfax, Sfax 3029, Tunisia; 5Department of Humanities, COMSATS University Islamabad, Islamabad Campus, Park Road, Islamabad 45550, Pakistan; drsukoon@gmail.com; 6High Institute of Sport and Physical Education of Sfax, University of Sfax, Sfax 3000, Tunisia; trabelsikhaled@gmail.com; 7Research Laboratory, Education, Motricity, Sport and Health, EM2S, LR19JS01, University of Sfax, Sfax 3000, Tunisia; 8Department of Nutrition and Food Technology, School of Agriculture, The University of Jordan, Amman 11942, Jordan; h.ghazzawi@ju.edu.jo; 9Division of Research and Development, Lovely Professional University, Phagwara 144411, Punjab, India; pandiperumal2023@gmail.com; 10Saveetha Medical College and Hospitals, Saveetha Institute of Medical and Technical Sciences, Saveetha University, Chennai 602105, Tamil Nadu, India; 11Department of Psychiatry, University of Toronto, Toronto, ON M5S, Canada; mary.seeman@utoronto.ca

**Keywords:** dietary assessment, dietary intake, mental health, nutritional evaluation, nutritional status, serious mental disorders

## Abstract

General food frequency questionnaires (FFQs) have not been tailored for or validated in individuals with psychiatric disorders. Given the unique eating behaviors of patients with serious mental illnesses (SMIs), custom-made tools are needed. Therefore, we developed and validated an FFQ customized to individuals with SMIs. A total of 150 adults with SMIs (schizophrenia, bipolar disorder, or major depression) were recruited from Bahrain. The participants completed the 50-item Dietary Intake Evaluation Questionnaire for Serious Mental Illness (DIETQ-SMI) FFQ and a 3-day food record (FR). The validity of the FFQ was assessed by comparing nutrient intake to FR intake using correlation and reliability statistics. The DIETQ-SMI demonstrated a good ranking validity compared to the FR based on correlation coefficients (rho 0.33 to 0.92) for energy and macro- and micronutrients. The FFQ had high internal consistency (McDonald’s omega = 0.84; Cronbach’s alpha = 0.91) and test–retest reliability (ICC > 0.90). The FFQ tended to estimate higher absolute intakes than the FR but adequately ranked the intakes. The FFQ value was correlated with the FR for all the items (*p* < 0.001). The DIETQ-SMI is a valid and reliable FFQ for ranking dietary intake in individuals with an SMI. It can help assess nutritional status and, subsequently, guide interventions in high-risk psychiatric populations.

## 1. Introduction

While the concept of serious mental illness (SMI) is widely used across research, practice, and policy settings, there is no consensus on its precise definition [1]. The term SMI refers to a severe mental illness and comprises a group of mental health disorders whose effects significantly undermine function. These disorders are chronic, meaning that they persist and require treatment over long periods of time [2]. The term encompasses diagnoses such as major depressive disorder, schizophrenia, and bipolar disorder [3], which are the focus of the present research. Patients with an SMI are at high risk for both nutritional deficiencies and obesity [4,5,6,7,8,9]. These issues arise due to the complex interplay between symptoms, medications used in the treatment, socioeconomic disadvantages faced by most patients with an SMI, and unhealthy lifestyles (social isolation, substance use, sedentariness) that frequently develop over time [5,6,7,8,10].

Psychiatric symptoms such as paranoia, anhedonia, low motivation, social withdrawal with consequent isolation, and absence of support can make shopping for and preparing nutritious meals challenging [5,6,7,8,11,12]. The antipsychotic medications used to manage psychosis and mood symptoms can increase appetite and lead to significant weight gain and further metabolic disturbances over time [13]. Loss of appetite due to comorbid depression can also occur [13], while psychiatric medications can increase nutritional problems through their impact on the absorption and metabolism of nutrients [13].

Many patients with SMIs live on restricted food budgets and have limited access to healthy food [5,6,7,8]. The causes for this include unemployment, alienation from family, residence in disadvantaged neighborhoods, and cognitive disability [5,6,7,8]. As a result, individuals with an SMI exhibit 2–3 times greater rates of obesity than the general population, with a subsequently increased risk of cardiovascular disease, diabetes, dyslipidemia, and other chronic health problems [14,15].

Deficiencies in important nutrients, such as β-carotene, folate, vitamin D, and vitamin B12, are common in this population [14,16]. This is due to a poor diet quality, metabolic changes resulting from medications, and inadequate exposure to sunlight [14,16].

Lifestyle interventions to improve diet and increase physical activity are critical to the nutritional needs of patients with SMIs [17,18,19]. Supporting individuals in overcoming socioeconomic barriers and managing the adverse metabolic effects of needed medications are also vital [20,21]. Hence, it is necessary to have a full and clear understanding of psychiatric patients’ nutritional status to develop a well-planned treatment.

An adequate nutritional status is essential for the maintenance of both physical and mental health [22]. However, valid and reliable dietary assessment instruments specific to this population are hard to find [14]. In other areas of clinical medicine and research, food frequency questionnaires (FFQs) are commonly used for long-term diet assessments [23]. Such questionnaires have proven useful because they are inexpensive and easy to administer [23].

While general food FFQs have been widely used in nutrition research, these standard instruments have never been designed specifically for individuals with psychiatric disorders [22]. Given the unique dietary behaviors and needs of individuals with an SMI, tailored and validated tools are needed to understand the dietary deficits of this population [17,18,19]. Several FFQs have been created for patients with an SMI [14,16]. However, these instruments focused solely on single disorders such as depression or schizophrenia. However, studies have not taken a broader view of assessing dietary intake across the spectrum of serious mental illnesses. Furthermore, the existing questionnaires have been tested on small sample sizes and lack psychometric validation using data from diverse demographic groups [23,24].

This approach is important because the unique challenges faced by individuals with schizophrenia, bipolar affective disorder, and other related severe disorders are known to impact their dietary behaviors and nutritional status in unique ways [18,25,26]. As previously mentioned, specific psychiatric symptoms, specific medication side effects, cognitive deficits, substance abuse, including caffeine and nicotine (tobacco smoking), the neighborhood in which one lives, and social isolation can all influence food choices and nutrient intake [14,27,28].

Therefore, we considered that an FFQ designed specifically for assessing the diets of patients with major mental illnesses was needed. We hope that this study will provide clinicians and researchers with an important tool for evaluating nutritional status, identifying potential nutrient deficiencies or excesses, and guiding targeted nutritional interventions. The aim of this study was to develop and validate an FFQ for practical dietary assessment in individuals with an SMI. Our FFQ is named the Dietary Intake Evaluation Questionnaire for Serious Mental Illness (DIETQ-SMI). We hypothesized that the DIETQ-SMI would demonstrate good validity and reliability for assessing dietary intake in this population. We expected the nutrient intake estimates from the DIETQ-SMI to show a significant positive correlation with the individual recollections of one’s diet over 3 days collected during the same time period. We also hypothesized that the DIETQ-SMI would demonstrate a good test–retest reliability when readministered within a 2-week period.

Our recruitment base was the SMI population of Bahrain. The country’s fiscal health is excellent, and mealtimes are often shared communally, with families and friends gathering to eat together [14]. Therefore, there should be reduced barriers to healthy eating for individuals with SMIs. However, globalization has had an influence on this, especially in urban areas [14]. Because they were not previously needed, Bahrain has no nationwide system of soup kitchens dedicated specifically to supporting people with SMIs. This may leave some patients to their own devices and highlights an area where community services could be developed to meet the nutritional and social needs of this population.

## 2. Materials and Methods

### 2.1. Development of the Food Frequency Questionnaire (DIETQ-SMI)

This longitudinal questionnaire validation study was conducted at the Psychiatric Hospital in Bahrain. The research protocol adhered to international guidelines for questionnaire design and followed a systematic multistep development process. Specifically, we followed the principles outlined in the guidelines for best practice in questionnaire design by the Survey Research Center at the University of Michigan, USA [29]. These guidelines provide a comprehensive framework for developing culturally appropriate and linguistically equivalent survey instruments, ensuring high-quality data collection across diverse populations. Furthermore, the research team adopted a systematic multistep approach to the development process, as recommended by Cade et al. [30] in their seminal work on constructing FFQs. This rigorous methodology involved the following key steps: Step 1: conducting a comprehensive literature review to identify dietary patterns, food preferences, and the unique nutritional challenges faced by individuals with SMIs. Step 2: consulting with a panel of experts, including psychiatrists, nutritionists, and researchers specializing in SMIs, to ensure the relevance and appropriateness of the food items and portion sizes included in the questionnaire. Step 3: developing an initial draft of the FFQ based on the information gathered from the literature review, expert panel, and qualitative data. Step 4: conducting cognitive interviews or pilot testing with a representative sample of the target population to assess the comprehensibility, clarity, and cultural relevance of the questionnaire items. Step 5: refining and finalizing the FFQ based on the feedback obtained during the pilot testing phase, ensuring that the instrument accurately captures the unique dietary patterns and food choices of a majority of individuals with SMIs.

A comprehensive list of foods and beverages commonly consumed by patients with SMIs was created by reviewing international/national dietary surveys, systematic reviews, and meta-analyses and consulting nutrition experts [16,23,31,32,33,34,35,36]. This initial food list contained more than 350 variants of individual food and beverage items. The list was organized into eight categories: grains (GR), fruits (FR), vegetables (VEG), dairy (DRY), meat and proteins (MP), oils and fats (OFI), beverages (BEV), snacks and sweets (SSS), and condiments (COND). The complete list of items is shown in Appendix A. The complete list of food and beverage items provided in Appendix A was developed through an exhaustive review of the existing dietary literature and consultations with experts.

To address the unique dietary needs and challenges of individuals with SMIs, the DIETQ-SMI was designed to include food items and portion sizes that reflected the typical eating behaviors of individuals with an SMI. In addition to the standard food categories found in general FFQs, this instrument incorporated items commonly consumed by individuals with SMIs, such as highly processed snacks (e.g., chips, cookies, candies, and ready-to-eat meals), fast foods (e.g., burgers, pizzas), and sugar-sweetened beverages (e.g., sugary drinks like sodas, juices, and energy drinks). Furthermore, the DIETQ-SMI included detailed questions about portion sizes and frequency of consumption for these items, as individuals with SMIs might have some difficulties with portion control and might engage in binge eating or emotional eating. By capturing these nuances, the DIETQ-SMI aimed to provide a more accurate assessment of nutrient intake and dietary patterns in this vulnerable population, facilitating targeted nutritional interventions and improving the overall health outcomes.

To reduce respondent burden, the number of FFQ items was reduced by eliminating rarely eaten foods, grouping similar foods together, and collapsing questions when appropriate; in our case (i.e., Bahrain), items not consumed for religious reasons were excluded (e.g., pork products). This process resulted in a 50-item FFQ with 10 supplementary questions on food preferences and behaviors. See Appendix A.

We also created skip patterns for items such as alcoholic beverages due to their prohibited use due to religious taboos. The standard portion sizes for each item were determined using data from national surveys. The recall period for consumption frequency was the preceding two weeks. The response options ranged from “never” to “2 or more times per day”. The full response options were “never/rarely”, “1–3 times/month”, “1–2 times/week”, “3–6 times/week”, “1 time/day”, and “2 times/day” [14]. The portion size of each food was determined based on the standard unit of servings or the midsize; e.g., an egg was assumed to be a midsize of 50 g (1.75 ounces (50 g) compared to a small one of 43 g (1.5 ounces (43 g) or a large one of 2 ounces (57 g (2 ounces)) [14]. The portion sizes were identified based on the local plate or bowl size. Thus, to further standardize portion estimation, food models and utensils were utilized. The participants were shown plates, bowls, glasses, and other servingware with examples of small, medium, and large portion sizes as a reference. The use of visual food models aligned with published recommendations for standardizing portion size estimation in food frequency questionnaires [14,16]. After determining the portion size, those who ate small portions were asked to respond with a value of 0.5, and those who ate relatively large portions were asked to respond with a value of 1.5. This approach enabled for the standardized quantification of the portion sizes consumed by the participants.

The final FFQ was evaluated for content validity by a panel of six psychiatric, nutritional, and research experts. The expert panel consisted of the core research team members, who each had more than 25 years of experience in their respective fields and held extensive knowledge in service delivery and research. The psychiatric experts HJ and ZS are based in Bahrain and are Arabic speakers. The nutritionist HG is based in Jordan, a neighboring Arabic-speaking country. KT and AA are experts based in Tunisia, another neighboring country in which Arabic is widely spoken. Additionally, the international experts SRP and MVS, with renowned expertise in psychiatry, served on the panel. Revisions were made based on their feedback before pilot testing the questionnaire on a sample of 25 participants from the target population. After pilot testing the FFQ with 20 people, final edits were made to improve clarity and readability. The pilot sample was deliberately excluded from the main analyses of this study to maintain the integrity and reliability of the findings.

### 2.2. Study Population and Settings

This study included a randomly selected sample of 150 adults aged 18–64 years with a mental illness living in the community in Bahrain. The study participants were recruited through the Rehabilitation Services Department of the Psychiatric Hospital, Bahrain. The Psychiatric Hospital in Bahrain, located in the capital city of Manama, is the country’s largest and only public facility that provides diagnosis and treatment for a wide range of mental illnesses [14]. With 300 beds, it is the sole public psychiatric hospital serving the country’s population of approximately 1.8 million people [14].

We included 50 patients diagnosed with depression (major depressive disorder, single episode, unspecified), 50 patients diagnosed with schizophrenia (any type), and 50 patients diagnosed with bipolar affective disorder (current episode manic without psychotic symptoms, current episode manic with psychotic symptoms, current episode mild or moderate depression, current episode severe depression without psychotic symptoms, current episode severe depression with psychotic symptoms, current episode mixed, currently in remission). All the diagnoses were made according to the *International Statistical Classification of Diseases, 10th Revision* by a multidisciplinary team (MDT) led by trained consultant adult psychiatrists [37]. This process of diagnosis by an MDT involved comprehensive clinical assessments, including interviews, observations, and, at times, the use of additional diagnostic tools for a more complete assessment. These procedures helped ensure accurate and reliable diagnoses in accordance with established diagnostic criteria.

### 2.3. Selection Criteria

The inclusion criteria for this study were as follows: [1] the patient had a confirmed diagnosis of an SMI; [2] they were currently receiving treatment; [3] they were aged 18 years or older; and [4] they were willing to participate in this study and signed a consent form. The exclusion criteria for the participants were [1] pregnancy/lactation, [2] following a specific diet, or [3] already participating in a clinical trial.

### 2.4. Procedures for Validation

The FFQ validation study was conducted against a 3-day dietary recall record (diary record) communicated through a mobile phone app. To validate the FFQ, patients with an SMI were recruited from November to December 2023. Once recruited, the participants were required to maintain a log of 3-day dietary records via an open-source app called “waistline”, available free from F-Droid, GitHub, etcetera [available at https://f-droid.org/en/packages/com.waist.line/, accessed on 1 September 2023].

A week later, all the participants were requested to complete a pencil-and-paper questionnaire at the clinic. The questionnaire included the following: anthropometric data; dietary habits; lifestyle questions; illness history; medications; dietary supplements; family history; psychotropic medication-induced side effects; FFQ estimates; and 3-day dietary records (two weekdays and one weekend). A random subset of the sample (n = 60) was asked to repeat the FFQ after two weeks to determine its test–retest reliability. This time interval was considered long enough to reduce memory effects but short enough to minimize true changes in dietary habits. We were careful to avoid collecting data during religious or cultural occasions when food intake is likely to deviate from an individual’s normal habits. For example, we ensured that the 3-day dietary records captured typical weekdays and weekend days, avoiding any records during the holy month of Ramadan when Muslims fast from dawn to sunset. The data collected from the FFQ and 3-day dietary records were reviewed by trained staff, and the integrity of the data was checked prior to entry.

### 2.5. Ethics

The study procedures were approved by the Research Committee of the Psychiatric Hospital, Bahrain (PREC/2023/1178). Written informed consent was obtained from all the participants prior to data collection, and all the procedures adhered strictly to the principles of the Declaration of Helsinki.

### 2.6. Data Preparation

Participant responses from the FFQ were entered into a statistical software database for analysis as a comma-separated value file. The frequency of consumption for each food item was converted into a daily equivalent intake. The total intakes of specific food groups and nutrients were calculated by summing the daily equivalents across the relevant items. Custom programming scripts were used to generate these intake estimates. The scripts utilized the nutritional values of the U.S. Department of Agriculture (available at https://fdc.nal.usda.gov/, accessed on 1 September 2023). For example, if a patient consumed a small apple (100 g), it was noted that it contained 52 calories, 0.3 g of protein, and 13.8 g of carbohydrates (2.4 g of fiber and 10 g of sugar). Its mineral content was calcium (6 mg), iron (0.12 mg), magnesium (5 mg), phosphorus (11 mg), and potassium (107 mg). The test mixture contained vitamin C (4.6 mg), thiamine (0.017 mg), riboflavin (0.026 mg), niacin (0.091 mg), vitamin B6 (0.041 mg), and folate (3 μg). Apples contained very little fat, saturated fat, or cholesterol (nil g) and small amounts of vitamin A (29 μg), vitamin E (0.18 mg), or vitamin K (2.2 μg).

The estimated energy requirement (EER) was calculated for each participant using equations published by the Institute of Medicine (IOM) [38]. These predictive equations estimate energy needs based on age, sex, weight, height, and physical activity level. Age, height, and weight were obtained from medical records. The physical activity level of each participant was estimated by the study dietitian based on self-reported usual activity. The equations for the EER differ for males and females because of differences in metabolism and body composition [38]. The appropriate equation was selected to estimate each participant’s energy needs based on their age, sex, weight status, and assigned activity level. The EER provides an estimate of the expected total energy expenditure and reflects the average dietary energy intake required to maintain caloric balance for a comparable healthy individual [38]. The estimated EER was used as a reference point to evaluate the plausibility of the reported energy intake from the FFQ and food records.

In this study, 21 parameters were computed. These included energy [kcal], protein [g], carbohydrate [g], total fat [g], saturated fat [g], monosaturated fat [g], polyunsaturated fat [g], fiber [g], cholesterol [mg], calcium [mg], iron [mg], magnesium [mg], sodium [mg], phosphorous [mg], potassium [mg], zinc [mg], niacin [mg], thiamine [mg], riboflavin [mg], vitamin B6 [mg], vitamin A [RE], and vitamin C [mg].

### 2.7. Data Analysis

Prior to analysis, the data were checked for normality via visualizations and formal tests, e.g., the Shapiro–Wilk test and the Kolmogorov–Smirnov test. Summary statistics, including medians and interquartile ranges (IQRs) and percentile distributions, were computed for all food groups and nutrients.

To evaluate the validity of the FFQ, the estimated intakes were compared to the 3-day dietary records. Correlation coefficients were used to assess the ability of the FFQ to accurately rank the participants’ habitual dietary intake [39]. Specifically, Wilcoxon signed-rank tests were used to compare nutrient intake between the two methods. Spearman’s rank correlation coefficient (rho) was computed to evaluate the correlation between the FFQ estimates and the food diet records [39].

The internal consistency of the FFQ was evaluated using McDonald’s omega [40] and Cronbach’s alpha [41]. Omega values ≥ 0.7 indicated a good reliability. The test–retest reliability of the FFQ was evaluated by having a subset of participants (n = 50) complete the FFQ a second time, approximately two weeks after the initial administration. The test–retest reliability was estimated using intraclass correlation coefficients (ICCs) between the nutrient and food group intakes from the two FFQ administrations [42]. ICCs were computed for both unadjusted intakes. ICC values above 0.40 were considered acceptable, with higher values indicating a better reproducibility. The reliability of the FFQ was also evaluated by estimating the proportion of participants classified into the same or adjacent quartile of intake on the two administrations [43]. Excellent reliability was defined as more than 50% of the participants being classified into the exact same quartile, and substantial reliability was defined as more than 50% of the participants being classified into the same or adjacent quartile in the repeat administration.

The Kruskal–Wallis test was used to compare the nutrient intake between the diagnostic groups (those with bipolar disorder, major depressive disorder, and schizophrenia spectrum disorders) for the food frequency questionnaire (FFQ) and the 3-day food records. The Kruskal–Wallis test is a nonparametric method for comparing two or more independent samples; this method is similar to the ANOVA but does not assume normality in the data. Since many nutrient intake variables had non-normal distributions based on the Kolmogorov–Smirnov test results, the Kruskal–Wallis test was chosen over the ANOVA to compare the diagnoses. When the Kruskal–Wallis test detected significant differences between the groups (*p* < 0.05), post hoc Dunn’s tests were performed to identify which diagnostic groups were different while controlling for multiple comparisons. The detailed results of the post hoc analyses are not reported in this paper. The Kruskal–Wallis test allowed for appropriate between-group comparisons of nutrient intake by psychiatric diagnosis from both the FFQ and food records despite non-normality in the data.

All the analyses were conducted using R for statistical computing (R version 4.3.2 [Eye Holes] was released on 31 October 2023). A *p* value < 0.05 was considered to indicate statistical significance.

## 3. Results

The questionnaire required approximately 15–20 min to complete.

The median age of the participants was 41.5 years (IQR 4 years), the median height was 166.5 cm (IQR 12 cm), the median weight was 72 kg (IQR 11 kg), and the median BMI was 26.23 kg/m^2^ (IQR 3.64 kg/m^2^). The participants were taking a median of three current psychotropic medications (IQR two medications) and had a median duration of illness of 7 years (IQR 4 years). The estimated energy requirement was 2055 kcal for female (IQR 138 kcal) and 2678 kcal for male participants (IQR 377 kcal). Of the 150 participants, 69 (46%) were male, and 129 (86%) were single. A family history of mental illness was reported in 76 (51%) participants. The most common psychiatric disorders were bipolar affective disorder in 50 (33.3%) participants, major depressive disorder in 50 (33.3%) participants, and schizophrenia and related disorders in 50 (33.3%) participants.

Energy intake was significantly greater in the FFQ method (median 2332.6 kcal, IQR 729.3 kcal) than in the FR method (median 2178.6 kcal, IQR 465.3 kcal, *p* = 0.001). Similar trends were observed for total fat, saturated fat, monounsaturated fat, polyunsaturated fat, cholesterol, calcium, magnesium, sodium, phosphorous, potassium, niacin, thiamin, riboflavin, vitamin A, and vitamin C, with the intakes from the FFQ significantly higher than those from the FR (all *p* < 0.05). In contrast, the median protein intake was significantly lower in the FFQ method (106.1 g, IQR 50.9 g) than in the FR method (124.1 g, IQR 34.3 g, *p* = 0.001). The median fiber intake was greater in the FFQ (22.7 g, IQR 25.7 g) than in the FR (29.4 g, IQR 15.6 g, *p* = 0.028). Iron intake was significantly lower in the FFQ (35.7 mg, IQR 28.1 mg) than in the FR (44.0 mg, IQR 24.7 mg, *p* = 0.003). No significant differences between the FFQ and FR were detected for carbohydrate, vitamin B6, or zinc intake (all *p* > 0.05). See Table 1.

Figure 1 compares the macro- and micronutrient estimates from the FFQ versus the FR. For the macronutrients, the FFQ overestimates energy, total fat, saturated fat, and monounsaturated fat compared to the FR, while it underestimates protein and fiber intake.

Regarding the micronutrients, the FFQ tends to overestimate cholesterol, calcium, sodium, magnesium, phosphorus, potassium, riboflavin, vitamin A, and vitamin C relative to the FR. In contrast, it underestimates iron and niacin. For some micronutrients such as zinc, thiamin, and vitamin B6, the two methods provide comparable estimates. In summary, the FFQ overestimates many macro- and micronutrients compared to the more accurate FR, while underestimating a few nutrients. This highlights the tendency for FFQs to over-report intake for a number of nutritional parameters, which should be considered when interpreting FFQ results.

The overall internal consistency of the DIETQ-SMI was high, with a McDonald’s ω of 0.84 (95% CI 0.80 to 0.88) and a standardized Cronbach’s α of 0.91 (95% CI 0.89 to 0.93). Most nutrients showed only minor changes in McDonald’s ω and Cronbach’s α when they were individually dropped from the analysis. The exceptions were sodium, which decreased McDonald’s ω to 0.80 when dropped, and potassium, which decreased it to 0.71. The item–rest correlations ranged from 0.23 for zinc to 0.73 for potassium. Higher correlations indicate that the nutrient contributes more to the overall internal consistency. Most nutrients had item–rest correlations between 0.40 and 0.70. See Table 2. The test–retest reliability exceeded 0.90 for all the nutritional items when the DIETQ-SM was repeated after two weeks to determine the test–retest reliability.

Statistically significant positive correlations were observed between the FFQ estimates and the food records for all the nutrients (all *p* < 0.001). The strongest correlations were observed for protein (rho = 0.91), zinc (rho = 0.92), and niacin (rho = 0.69), indicating excellent agreement between the two methods for these nutrients. Moderate correlations were observed for fiber, saturated fat, polyunsaturated fat, phosphorous, carbohydrate, cholesterol, sodium, riboflavin, and thiamin (rho between 0.49 and 0.61). Weaker but still statistically significant correlations were noted for the remaining nutrients, including energy, calcium, vitamin A, vitamin B6, and vitamin C (rho between 0.33 and 0.50). See Table 3.

For the FFQ, significant differences by diagnosis were found for niacin, riboflavin, vitamin A, and vitamin B6 (all *p* < 0.05). Those with bipolar affective disorder (BAD) had the highest intakes of these nutrients. According to the food records, significant differences according to diagnosis were detected for fiber, niacin, thiamine, riboflavin, vitamin A, vitamin B6, and vitamin C (all *p* < 0.05). Again, those with BAD generally had higher intakes, except for fiber and vitamin C, for which those with major depressive disorder had higher intakes. For most nutrients, there were no significant differences between psychiatric diagnoses according to either dietary assessment method. However, both methods detected greater intakes of several B vitamins and antioxidants, such as vitamin A, in patients with bipolar disorder than in those with schizophrenia spectrum disorders. The food records revealed more differences between diagnostic groups, likely because a more detailed nutrient assessment was provided over 3 days than what had been available for the FFQ, which estimated the usual intake over the past month. Nevertheless, the FFQ was able to detect some differences in nutrient intake by psychiatric diagnosis. See Table 4.

## 4. Discussion

The need for an FFQ tailored to individuals with SMIs arose from the unique dietary patterns and food choices observed in this population. Existing FFQs designed for the general population might not have adequately captured the dietary habits and nutrient intake profiles specific to individuals with mood and psychotic disorders. Research has shown that individuals with SMIs often have dietary imbalances, characterized by a higher intake of energy-dense nutrient-poor foods and a lower consumption of fruits, vegetables, and whole grains [5,6,7,8,10]. These dietary patterns could exacerbate metabolic complications and contribute to the higher prevalence of obesity, diabetes, and cardiovascular diseases observed in this population [14,44]. The DIETQ-SMI differs from existing FFQs for the general population in several key aspects. It includes an expanded list of highly processed energy-dense snack foods, convenience items, fast-food options, and sugar-sweetened beverages, reflecting the dietary patterns commonly observed in individuals with SMIs. While the food items themselves might not have necessarily had unique micronutrient profiles, the DIETQ-SMI was designed to better capture potential nutritional imbalances and deficiencies that were more prevalent in certain SMI populations due to their inability to afford nutritious food, disorder-induced apathy and lack of motivation to prepare nutritious food, use of substance that cut appetite, use of medications that induce a need for calorie-rich food, night eating, binge eating, and solitary eating [5,45,46].

The FFQ tended to estimate higher intakes of energy and most nutrients than the more detailed FR. Some exceptions were protein, fiber, and iron. The reliability of the results indicates that the DIETQ-SMI has a high internal consistency for assessing nutrient intake in this population. No single nutrient substantially lowered the internal reliability when dropped. These findings suggest the appropriate internal validity and factorial unity of the FFQ for ranking nutrient intake in this population. The results demonstrated the good ranking validity of the DIETQ-SMI compared to that of food records. The strongest agreement was observed for protein, zinc, and niacin. The FFQ showed moderate-to-weak correlations for other nutrients but was still able to significantly rank intakes when compared to the reference method. These findings indicate an acceptable relative validity for assessing nutrient intake with this FFQ. Both the FFQ and food records suggest that individuals with bipolar disorder have higher intakes of certain micronutrients, such as B vitamins and vitamin A, than patients with depression or schizophrenia spectrum disorders. The FFQ appears to be adequate for detecting some diagnostic differences in this population.

This study provides important evidence for the validity of a tailored FFQ for assessing dietary intake in people with severe mental illnesses. The DIETQ-SMI demonstrated a good internal consistency and an acceptable ranking validity against 3-day food records. These findings indicate that the FFQ can be used to appropriately rank the intake of energy and macro- and micronutrients in this specific population.

Our findings are consistent with the previous literature demonstrating challenges in dietary assessment among individuals with mental illnesses [16,23,47,48]. The high validity of our tailored FFQ addresses the limitations of existing tools and will help advance nutrition research in this population [23]. Evidence of distinct dietary patterns by diagnosis also aligns with prior studies showing that nutritional risks may differ between diagnostic groups [14]. The validity of our FFQ helps fill critical gaps in knowledge and practice capabilities regarding diet and mental health.

The validated DIETQ-SMI provides an important new tool for dietitians and clinicians working with psychiatric populations to assess dietary intake and nutritional status. Nutrition questions are not often asked during standard psychiatric interviews, whereas this FFQ can be easily incorporated into routine screening prior to appointments [49]. It highlights dietary risks, guides nutritional interventions, and is a research tool for examining diet–mental health associations. For example, findings from this study regarding the intake of B vitamins and antioxidants such as vitamin A being high in individuals with bipolar disorder relative to individuals with other SMIs may be clinically important but would need to be controlled at the income level because bipolar illness is associated with a higher socioeconomic status than, for instance, schizophrenia [50]. The FFQ also identified diet quality issues such as high sodium and low fiber intakes, highlighting key nutritional targets for counseling and behavioral change strategies in this population.

Similarly, a recent study revealed that 32% of inpatients and 34% of outpatients exhibited malnutrition, which, in turn, resulted in worsening mental symptoms [49]. According to the BMI of both groups, they were classified as overweight. Therefore, there is a need to develop a specialized screening tool for psychiatric patients to assess their nutritional needs [49]. However, further research is needed to validate the DIETQ-SMI in more diverse demographic groups to confirm its applicability across different ages, races/ethnicities, and socioeconomic statuses. This FFQ can be used to advance the understanding of diet–mental health interactions, including elucidating the effects of overall diet quality as well as specific nutrients on psychiatric outcomes. One question not examined here was the nutritional status of women with an SMI who are also parents. Such women often sacrifice their own nutrition to provide for their children, a behavior which can severely undermine their own health [51]. Long-term studies should examine the sensitivity of the FFQ to change to accurately assess nutritional interventions over time. Identifying biological mechanisms linking diet and mental illness relies on accurate dietary assessments, underscoring the value of this tailored FFQ for nutrition and psychiatry research. The associations found between dietary intake and mental illness diagnosis merit additional investigation to clarify whether dietary risks and their management need to differ substantially among individuals with psychiatric disorders.

A key strength of this study was the development of an FFQ specifically for individuals with mental illness, as preexisting FFQs may not capture dietary behaviors or food choices relevant to psychiatric populations. Tailoring the questionnaire to include foods commonly consumed by individuals with schizophrenia, bipolar disorder, or major depression likely improved the instrument’s validity. Other strengths included the use of statistical techniques such as McDonald’s omega and ICC analysis, which are robust for dietary validation studies.

Potential limitations should be noted. The use of 3-day food records as a reference may have introduced recall bias and the underestimation of intake. The participants completed the FFQ and food records in close proximity to each other, which could have led to correlated errors between the two instruments. The sample was predominantly composed of Arab adults under 65 years of age; thus, the findings may not be generalizable to other demographic groups. Eating habits, socioeconomic levels, and familial cohesion and support related to food may differ in various regions of the world, all of which may affect access to a healthy diet. Another potential limitation is the sample size of 150 participants split between three groups. This relatively small sample size may limit the generalizability of the findings and warrants caution in interpreting the results. Future research with larger sample sizes would be beneficial to further validate the findings and enhance the generalizability of this study’s conclusions. Finally, the state of illness and the influence of specific psychotropic medications on metabolism were not systematically captured in our study. These factors could potentially impact dietary patterns and nutrient intake among individuals with SMIs. Therefore, it is important to note this limitation in our research. Future studies should consider incorporating measures to systematically capture the state of illness and the specific psychotropic medications used by the participants. By doing so, a more comprehensive understanding of the relationship between dietary patterns, nutrient intake, and these influential factors can be achieved. This would enhance the validity and clinical applicability of the findings, providing valuable insights for tailored nutritional interventions in individuals with SMIs.

A growing body of research has demonstrated bidirectional relationships between diet quality and mental health outcomes [52]. For example, several studies indicate that diets high in processed foods, saturated fats, and refined carbohydrates can negatively impact mood and cognition, whereas diets rich in fruits, vegetables, whole grains, and omega-3 fatty acids are associated with a reduced risk of depression and other psychiatric disorders [5,10,53]. The proposed mechanisms for these effects include modulation of the gut microbiota, inflammation, oxidative stress, and neuroplasticity through dietary components that influence immune and metabolic pathways [52]. Further elucidating the relationship between diet and mental health in patients with serious mental illnesses may reveal modifiable risk factors and opportunities for nutritional interventions. This highlights the need, as a first step, for validated tools such as our tailored FFQ to accurately capture dietary habits.

It is important to interpret our study findings within the cultural and economic context of Bahrain. Compared to those in other countries, dietary deficiencies may be less prevalent among individuals with SMIs in Bahrain for several reasons. First, the average income is relatively high, which increases accessibility to nutritious foods. Second, strong family connectedness frequently provides social support, including shared household meals. Additionally, many patients with SMIs receive government disability benefits that are likely to facilitate adequate nutrition. Finally, alcohol consumption is prohibited in Bahrain, eliminating a factor which can detrimentally impact the diet among people with mental illnesses elsewhere. However, while nutritional deficiencies may be less common in Bahrain than in other nations, assessing dietary intake remains a priority for identifying poor eating habits and informing targeted interventions to optimize nutrition in this vulnerable group. Our validated DIETQ-SMI provides an important tool for rigorously evaluating dietary patterns among individuals with an SMI in the unique cultural context of Bahrain.

## 5. Conclusions

In this study, we developed and validated a food frequency questionnaire (DIETQ-SMI) to assess nutrient intake in people with severe mental illnesses. The DIETQ-SMI showed a high internal consistency and good agreement with 3-day food records for ranking intakes of energy, macronutrients, and micronutrients. The FFQ tended to estimate higher absolute intakes than the food records but had moderate-to-strong correlations in terms of ranking nutrient intakes. We also found that the FFQ could detect some differences in nutrient intake between diagnoses, specifically higher intakes of B vitamins and vitamin A in patients with bipolar disorder. The DIETQ-SMI demonstrated satisfactory validity and reliability for assessing dietary intake in this population. The FFQ can be considered an appropriate dietary assessment tool for ranking nutrient intake in people with severe mental illnesses. Its specificity can be modified in different parts of the world. Some caution is needed in interpreting absolute intakes derived from the FFQ, as it may overestimate intakes compared to food records. Further research can build upon these findings to elucidate nutritional gaps relevant to mental health outcomes in culturally specific psychiatric populations.

## Figures and Tables

**Figure 1 brainsci-14-00312-f001:**
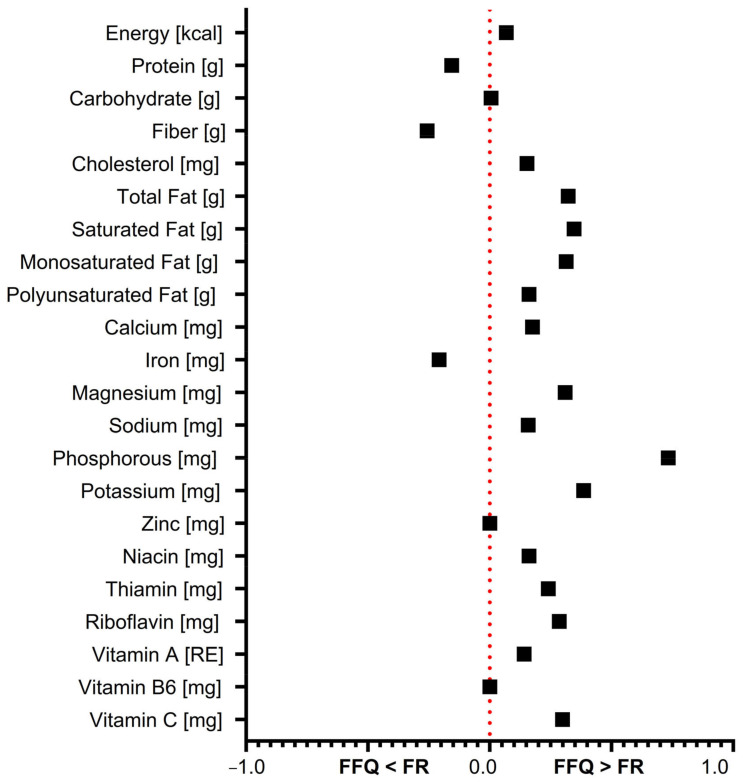
The difference between nutrient estimates from the FFQ versus the FR. Notes: Positive values indicate that the FFQ estimated a higher intake than the FR. Negative values indicate that the FFQ estimated a lower intake. Values near zero indicate similar estimates between the two methods.

**Table 1 brainsci-14-00312-t001:** Differences between the DIETQ-SMI FFQ and 3-day food records.

Variable	FFQ *		3-Day Food Record **		Direction	*p* Value
**Energy [kcal]**	2332.6	729.3	2178.6	465.3	FFQ > FR	0.001
**Protein [g]**	106.1	50.9	124.1	34.3	FFQ < FR	0.001
**Carbohydrate [g]**	301.9	100.9	300.2	64.1	FFQ = FR	0.278
**Fiber [g]**	22.7	25.7	29.4	15.6	FFQ > FR	0.028
**Cholesterol [mg]**	262.2	176.5	225.0	150.6	FFQ > FR	0.001
**Total Fat [g]**	80.7	32.8	58.3	23.8	FFQ > FR	0.001
**Saturated Fat [g]**	32.2	13.2	22.7	7.3	FFQ > FR	0.001
**Monosaturated Fat [g]**	25.8	11.0	18.8	8.6	FFQ > FR	0.001
**Polyunsaturated Fat [g]**	20.1	13.1	17.1	9.0	FFQ > FR	0.001
**Calcium [mg]**	648.8	408.6	544.3	255.5	FFQ > FR	0.001
**Iron [mg]**	35.7	28.1	44.0	24.7	FFQ > FR	0.003
**Magnesium [mg]**	409.6	211.6	299.8	174.2	FFQ > FR	0.001
**Sodium [mg]**	2305.9	1319.6	1968.5	961.0	FFQ > FR	0.001
**Phosphorous [mg]**	915.5	357.6	424.3	202.2	FFQ > FR	0.001
**Potassium [mg]**	3862.1	1562.4	2614.2	1326.0	FFQ > FR	0.001
**Zinc [mg]**	8.4	5.2	8.4	3.5	FFQ > FR	0.082
**Niacin [mg]**	23.4	11.6	19.9	9.4	FFQ > FR	0.001
**Thiamin [mg]**	1.4	0.6	1.1	0.6	FFQ > FR	0.001
**Riboflavin [mg]**	1.6	0.6	1.2	0.5	FFQ > FR	0.001
**Vitamin A [RE]**	938.8	700.0	815.1	378.1	FFQ > FR	0.001
**Vitamin B6 [mg]**	2.6	1.0	2.6	0.9	FFQ = FR	0.455
**Vitamin C [mg]**	80.4	55.3	59.5	36.4	FFQ > FR	0.001

**Notes**: Data are expressed as medians and interquartile ranges (IQRs). * = Food frequency questionnaire; ** = 3-day food record.

**Table 2 brainsci-14-00312-t002:** Internal consistency of the DIETQ-SMI-FFQ.

Item	McDonald’s ω	Cronbach’s α *	Item–Rest Correlation
DIETQ-SMI FFQ	0.84 (95% CI 0.80–0.88)	0.91 (95% CI 0.89–0.93)	N/A
If item dropped			
Energy [kcal]	0.85	0.90	0.68
Protein [g]	0.84	0.91	0.60
Carbohydrate [g]	0.84	0.91	0.42
Fiber [g]	0.84	0.91	0.50
Cholesterol [mg]	0.84	0.91	0.46
Total Fat [g]	0.84	0.91	0.62
Saturated Fat [g]	0.84	0.90	0.62
Monosaturated Fat [g]	0.84	0.91	0.48
Polyunsaturated Fat [g]	0.84	0.91	0.48
Calcium [mg]	0.84	0.91	0.43
Iron [mg]	0.84	0.91	0.64
Magnesium [mg]	0.84	0.91	0.40
Sodium [mg]	0.80	0.91	0.57
Phosphorous [mg]	0.84	0.90	0.47
Potassium [mg]	0.71	0.91	0.73
Zinc [mg]	0.84	0.91	0.23
Niacin [mg]	0.84	0.91	0.36
Thiamin [mg]	0.84	0.91	0.31
Riboflavin [mg]	0.84	0.91	0.49
Vitamin A [RE]	0.86	0.91	0.30
Vitamin B6 [mg]	0.84	0.90	0.70
Vitamin C [mg]	0.84	0.91	0.42

Notes: * = Standardized Cronbach’s α.

**Table 3 brainsci-14-00312-t003:** Correlation coefficient between the DIETQ-SMI FFQ estimates and the 3-day food records.

Variable	Spearman’s Rho	*p* Value
Energy [kcal]	0.48	0.001
Protein [g]	0.91	0.001
Carbohydrate [g]	0.59	0.001
Fiber [g]	0.72	0.001
Cholesterol [mg]	0.59	0.001
Total Fat [g]	0.50	0.001
Saturated Fat [g]	0.60	0.001
Monosaturated Fat [g]	0.52	0.001
Polyunsaturated Fat [g]	0.61	0.001
Calcium [mg]	0.50	0.001
Iron [mg]	0.33	0.001
Magnesium [mg]	0.52	0.001
Sodium [mg]	0.52	0.001
Phosphorous [mg]	0.57	0.001
Potassium [mg]	0.50	0.001
Zinc [mg]	0.92	0.001
Niacin [mg]	0.69	0.001
Thiamin [mg]	0.49	0.001
Riboflavin [mg]	0.49	0.001
Vitamin A [RE]	0.41	0.001
Vitamin B6 [mg]	0.38	0.001
Vitamin C [mg]	0.53	0.001

**Table 4 brainsci-14-00312-t004:** Differences in dietary assessment methods based on the diagnosis of SMI.

Variable	Diagnosis	FFQ	*p* Value	FR	*p* Value
Energy [kcal]	BAD	2341.9 (737.6)	0.32	2280.3 (457.1)	0.07
	MDD	2492.9 (744.8)		2197 (319)	
	SD	2214.4 (613.7)		2090.5 (368.1)	
Protein [g]	BAD	110.1 (57.8)	0.43	123.7 (39.7)	0.40
	MDD	109.9 (45.5)		128 (33.1)	
	SD	101.5 (46.4)		121.1 (23.3)	
Carbohydrate [g]	BAD	300.1 (96.3)	0.25	309.5 (59.7)	0.13
	MDD	321.6 (96.1)		302.2 (52)	
	SD	300.4 (95.1)		288.7 (70.5)	
Fiber [g]	BAD	22.1 (25.4)	0.08	30.2 (14)	0.03
	MDD	25.5 (31.7)		31.2 (16)	
	SD	20.6 (14.8)		25.2 (13.3)	
Cholesterol [mg]	BAD	247.8 (172.4)	0.72	265 (182.5)	0.18
	MDD	270.3 (160.6)		216 (129.6)	
	SD	268.4 (182.9)		214.7 (107)	
Total Fat [g]	BAD	80.7 (29.4)	0.77	65.1 (22)	0.33
	MDD	82.2 (34.5)		58.3 (21.2)	
	SD	77.9 (34)		54.7 (25.4)	
Saturate Fat [g]	BAD	31.9 (8.4)	0.84	23.6 (6.8)	0.41
	MDD	33 (13.2)		22.4 (6.3)	
	SD	32.3 (16)		22.1 (7.1)	
Monosaturated Fat [g]	BAD	25.3 (9)	0.56	21.1 (8.6)	0.20
	MDD	27.5 (10.2)		18.5 (8.1)	
	SD	25.1 (11.5)		17.2 (8.5)	
Polyunsaturated Fat [g]	BAD	20.2 (12.1)	0.89	18.1 (8.6)	0.27
	MDD	19.6 (16.6)		16.9 (8)	
	SD	21.1 (11.9)		15.5 (10.1)	
Calcium [mg]	BAD	633.9 (297.3)	0.39	583.7 (273.8)	0.11
	MDD	682.5 (390.2)		543 (210.2)	
	SD	602.5 (499.7)		508.5 (284.8)	
Iron [mg]	BAD	35.9 (28.8)	0.26	50 (28)	0.06
	MDD	38.6 (24.3)		43.7 (16)	
	SD	33.5 (25.1)		37.3 (22.5)	
Magnesium [mg]	BAD	404 (248.7)	0.06	321.8 (220.3)	0.09
	MDD	446.3 (130.9)		315.5 (115.2)	
	SD	382.6 (175.7)		260.3 (165.5)	
Sodium [mg]	BAD	2151.9 (1445.5)	0.05	2160.7 (938.3)	0.05
	MDD	2610.4 (1176.1)		1964.1 (820.8)	
	SD	2194.4 (790.1)		1893.9 (890.9)	
Phosphorous [mg]	BAD	905.4 (288.3)	0.11	481.8 (211.7)	0.08
	MDD	948.4 (330.1)		416.7 (188.6)	
	SD	806.2 (378)		396.3 (142.6)	
Potassium [mg]	BAD	3790 (1343.4)	0.13	2924.1 (1426)	0.05
	MDD	4127.2 (1459.1)		2743.6 (1061)	
	SD	3539.2 (1685.1)		2326.4 (1436.9)	
Zinc [mg]	BAD	8.2 (3.5)	0.31	8.5 (3.3)	0.15
	MDD	8.6 (5.4)		9.1 (3.9)	
	SD	7.6 (5.1)		7.9 (3.2)	
Niacin [mg]	BAD	25.1 (11.3)	0.04	21.4 (10.9)	0.04
	MDD	25.6 (11.5)		20.6 (7.4)	
	SD	21 (8.6)		17.5 (6.9)	
Thiamin [mg]	BAD	1.4 (0.5)	0.13	1.2 (0.7)	0.05
	MDD	1.5 (0.6)		1.1 (0.4)	
	SD	1.3 (0.6)		1 (0.4)	
Riboflavin [mg]	BAD	1.6 (0.7)	0.03	1.3 (0.7)	0.01
	MDD	1.7 (0.5)		1.2 (0.5)	
	SD	1.4 (0.6)		1.1 (0.5)	
Vitamin A [RE]	BAD	1004.9 (781.4)	0.03	881.4 (380.7)	0.01
	MDD	1010.6 (573.1)		841.6 (315.1)	
	SD	783.4 (583.8)		717.8 (332.6)	
Vitamin B6 [mg]	BAD	2.6 (1)	0.001	2.9 (1.1)	0.02
	MDD	2.8 (1)		2.7 (0.7)	
	SD	2.3 (0.9)		2.4 (0.8)	
Vitamin C [mg]	BAD	86.1 (53.6)	0.28	68.1 (35.8)	0.04
	MDD	73.3 (57.9)		56.9 (32.2)	
	SD	82.3 (45.9)		54.3 (36.2)	

Notes: BAD = bipolar affective disorder; MDD = major depressive disorder; and SD = schizophrenia and related disorders.

## Data Availability

The data that support the findings of this study are available from the corresponding author (Haitham Jahrami) upon request. The data are not publicly available due to [ethical restriction].

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
