# Peer review of "Development and Validation of a Food Frequency Questionnaire for Evaluating the Nutritional Status of Patients with Serious Mental Illnesses (DIETQ-SMI) in Bahrain"

_brainsci, 2024, doi:10.3390/brainsci14040312_

Round 1
Reviewer 1 Report
Comments and Suggestions for Authors
This is a very interesting and well-written study regarding the development of a questionnaire to access eating habits of individuals with SMI. I have few major and some minor concerns to help elucidate it better. I hope authors find it useful.
Title: I suggest adding clearly where the study has been conducted (Bahrain), due to different cultural backgrounds that it might emerge.
Introduction:
- The concept of Serious mental Illness (SMI) is not a consensus and should be at least referenced.
- Lines 53-56: the two sentences are redundant.
- It is not clear for me if there are more questionnaires developed to this population, if so, what are their gaps and why is needed to developed one more.
Methods:
- I would move the “understanding culture” to the introduction, when talking about the cultural aspects and introducing that the study has been carried in Bahrain.
- Please provide a general paragraph before the “development…” to explain the study design, where it has been conducted, if you followed any specific guideline, and the ethical approval.
- It is very important to your method to clearly state where are these experts from? How many?
- I suggest creating a section regarding other independent variables as well as how it was calculated and drop it from the statistical analysis. Maintain the statistical part only to those related to the primary/secondary outcomes.
Results:
- Table 1 is too simple and should be dropped and the information moved to the text.
- Please state the duration of questionnaire’s response.
Discussion:
- The two first sentences are redundant.

Author Response
This is a very interesting and well-written study regarding the development of a questionnaire to access eating habits of individuals with SMI. I have few major and some minor concerns to help elucidate it better. I hope authors find it useful.
Authors’ response: Thank you for your thoughtful review of our manuscript and for providing constructive feedback. We appreciate you taking the time to thoroughly evaluate our work and provide suggestions to improve it. We have carefully considered all of the comments you raised during the peer review process. In response, we have made revisions and highlighted them in yellow for ease of re-review.
English language was further polished by the last author Prof. Mary Seeman an internationally recognized psychiatrists who is native speaker.
Title: I suggest adding clearly where the study has been conducted (Bahrain), due to different cultural backgrounds that it might emerge.
Authors’ response: We agreed with the reviewer and thus added location where the study has been conducted i.e., Bahrain as follow: “Development and Validation of a Food Frequency Questionnaire for Evaluating the Nutritional Status of Patients with Serious Mental Illness (DIETQ-SMI) in Bahrain”.
Introduction:
- The concept of Serious mental Illness (SMI) is not a consensus and should be at least referenced.
Authors’ response: We agree with the reviewer thus we explained the following: “The concept of serious mental illness (SMI) is widely used across research, practice, and policy settings; however, there is no consensus on its precise definition [1]. SMI, also known as severe mental illnesses, and are a group of mental health disorders that significantly impact a person's function. These disorders are typically chronic, meaning they persist over a long period of time, and they often require ongoing treatment and management [2]. SMI includes major depressive disorder, schizophrenia and bipolar disorder [3], which are the focus of the present research”.
We added the following references:
- Gonzales, L., L. E. Kois, C. Chen, L. López-Aybar, B. McCullough and K. J. McLaughlin. "Reliability of the term "serious mental illness": A systematic review." Psychiatr Serv 73 (2022): 1255-62. 10.1176/appi.ps.202100661.
- Baranyi, G., S. Fazel, S. D. Langerfeldt and A. P. Mundt. "The prevalence of comorbid serious mental illnesses and substance use disorders in prison populations: A systematic review and meta-analysis." The Lancet Public Health 7 (2022): e557-e68. 10.1016/S2468-2667(22)00093-7. https://doi.org/10.1016/S2468-2667(22)00093-7.
- Lim, C. T., M. P. Caan, C. H. Kim, C. M. Chow, H. S. Leff and M. C. Tepper. "Care management for serious mental illness: A systematic review and meta-analysis." Psychiatr Serv 73 (2022): 180-87. 10.1176/appi.ps.202000473.
- Lines 53-56: the two sentences are redundant.
Authors’ response: We agree with the reviewer thus we deleted the redundant sentence: “These symptoms include paranoia, low motivation, social withdrawal, and cognitive deficits”.
- It is not clear for me if there are more questionnaires developed to this population, if so, what are their gaps and why is needed to developed one more.
Authors’ response: Thank you for raising this important point about the need to justify developing a new food frequency questionnaire for individuals with serious mental illness. We appreciate you highlighting this gap in the introduction. In response, we have expanded the introduction to provide more context about existing food frequency questionnaires and their limitations for the SMI population as follow: “While general food FFQs have been widely used in nutrition research, these standard instruments were not designed specifically for individuals with psychiatric disorders [22]. Given the unique dietary behaviors and needs of those with SMI, tailored and validated tools are required to accurately capture the eating habits of this population [17-19]. To date, only a few FFQs have been created for those with SMI [14, 16]. However, these instruments focused solely on singular disorders like depression or schizophrenia. They have not taken a broader view to assess dietary intake across the spectrum of serious mental illnesses. Furthermore, existing questionnaires have been limited by small sample sizes and lacked psychometric validation using data from diverse demographic groups [23, 24]”.
Methods:
- I would move the “understanding culture” to the introduction, when talking about the cultural aspects and introducing that the study has been carried in Bahrain.
Authors’ response: We agreed with the reviewer. We moved the paragraph/section “understanding culture” to the introduction.
- Please provide a general paragraph before the “development…” to explain the study design, where it has been conducted, if you followed any specific guideline, and the ethical approval.
Authors’ response: We added the following: “This cross-sectional questionnaire validation study was conducted at the Psychiatric Hospital, Bahrain. The research protocol adhered to international guidelines for questionnaire design and followed a systematic multi-step development process”.
- It is very important to your method to clearly state where are these experts from? How many?
Authors’ response: We added the following: “The expert panel consisted of the core research team members, who each have over 25 years of experience in their respective fields and hold extensive knowledge in service de-livery and research. The psychiatric experts HJ and ZS are based in Bahrain and are Arabic speakers. The nutritionist HG is based in Jordan, a neighboring Arabic-speaking country. Experts KT and AA are based in Tunisia, another neighboring country where Arabic is widely spoken. Additionally, international experts SRP and MVS, with renowned expertise in psychiatry, served on the panel”.
- I suggest creating a section regarding other independent variables as well as how it was calculated and drop it from the statistical analysis. Maintain the statistical part only to those related to the primary/secondary outcomes.
Authors’ response: We agreed with the reviewer we created a section called 2.6. Data preparation regarding other independent variables as well as how it was calculated and dropped it from the statistical analysis. We only kept data analyses in the section labelled 2.7. Data analysis.
2.6. Data preparation
Participant responses from the FFQ were entered into a statistical software database for analysis as a comma separated value file. The frequency of consumption for each food item was converted into daily equivalent intake. Total intakes of specific food groups and nutrients were calculated by summing the daily equivalents across relevant items. Custom programming scripts were used to generate these intake estimates. The scripts utilized the nutrition values of the U.S. Department of Agriculture [available https://fdc.nal.usda.gov/]. To illustrate, if a patient consumed a small apple (100 g) this was noted as containing 52 calories and 0.3g protein, 13.8g carbohydrates (2.4g fiber and 10g sugar). Its mineral content was calcium (6mg), iron (0.12mg), magnesium (5mg), phosphorus (11mg), and potassium (107mg). It contained vitamin C (4.6mg), along with thiamine (0.017mg), riboflavin (0.026mg), niacin (0.091mg), vitamin B6 (0.041mg), and folate (3μg). The apple was noted as containing very little fat, saturated fat, or cholesterol (nil g) and small amounts of vitamin A (29μg), vitamin E (0.18mg), and vitamin K (2.2μg).
Estimated energy requirement (EER) was calculated for each participant, using equations published by the Institute of Medicine (IOM) [36]. These predictive equations estimate energy needs based on age, sex, weight, height and physical activity level. Age, height and weight were obtained from medical records. Physical activity level was estimated for each participant by the study dietitian, based on self-reported usual activity. The equations for EER differ for males and females because of differences in metabolism and body composition [36]. The appropriate equation was selected to estimate each participant’s energy needs based on their age, sex, weight status, and assigned activity level. The EER provides an estimate of expected total energy expenditure and reflects the average dietary energy intake required to maintain caloric balance for a comparable healthy individual [36]. The estimated EER was used as a reference point to evaluate the plausibility of reported energy intakes from the FFQ and food records.
In this study, 21 parameters were computed. These included energy [kcal], protein [g], carbohydrate [g], total fat [g], saturated fat [g], monosaturated fat [g], polyunsaturated fat [g], fiber [g], cholesterol [mg], calcium [mg], iron [mg], magnesium [mg], sodium [mg], phosphorous [mg], potassium [mg], zinc [mg], niacin [mg], thiamine [mg], riboflavin [mg], vitamin B6 [mg], vitamin A [RE], and vitamin C [mg].
2.7. Data analysis
Prior to analysis, the data were checked for normality via visualizations and formal tests, e.g., the Shapiro‒Wilk test and Kolmogorov‒Smirnov test. Summary statistics, including medians and interquartile ranges (IQRs) and percentile distributions, were computed for all food groups and nutrients.
To evaluate the validity of the FFQ, estimated intakes were compared to 3day dietary records. Correlation coefficients were used to assess the ability of the FFQ to accurately rank participants' habitual dietary intake. Wilcoxon signed rank tests were used to compare nutrient intake between the two methods. Pearson’s correlation coefficients were computed to evaluate the correlation between the FFQ score and food diet records.
The internal consistency of the FFQ was evaluated using McDonald's omega and Cronbach's alpha. Omega values ≥0.7 indicate good reliability. The test-retest reliability of the FFQ was evaluated by having a subset of participants (n = 50) complete the FFQ a second time, approximately two weeks after initial administration. Test-retest reliability was estimated using intraclass correlation coefficients (ICCs) between nutrient and food group intakes from the two FFQ administrations. ICCs were computed for both unadjusted intakes. ICC values above 0.40 were considered acceptable, with higher values indicating better reproducibility. The reliability of the FFQ was also evaluated by estimating the proportion of participants classified into the same or adjacent quartile of intake on the two administrations. Excellent reliability was defined as more than 50% of participants classified into the exact same quartile, and substantial reliability was defined as more than 50% classified into the same or adjacent quartile on the repeat administration.
The Kruskal‒Wallis test was used to compare nutrient intake between diagnostic groups (those with bipolar disorder, major depressive disorder, and schizophrenia spectrum disorders) for the food frequency questionnaire (FFQ) and 3day food records. The Kruskal‒Wallis test is a nonparametric method for comparing two or more independent samples; this method is similar to ANOVA but does not assume normality in the data. Since many nutrient intake variables had nonnormal distributions based on Kolmogorov‒Smirnov test results, the Kruskal‒Wallis test was chosen over ANOVA to compare diagnoses. When the Kruskal‒Wallis test detected significant differences between groups (p < 0.05), post hoc Dunn's tests were performed to identify which diagnostic groups were different while controlling for multiple comparisons. Detailed results of post hoc analyses are not reported in this paper. The Kruskal‒Wallis test allowed appropriate between group comparisons of nutrient intake by psychiatric diagnosis from both the FFQ and food records despite nonnormality in the data.
All the analyses were conducted using R for statistical computing (R version 4.3.2 [Eye Holes] was released on 20231031). A p value < 0.05 was considered to indicate statistical significance.
Results:
- Table 1 is too simple and should be dropped and the information moved to the text.
Authors’ response: We agree with the reviewer. We provided textual information about Table 1 in the results as follow: “The median age of the participants was 41.5 years (IQR 4 years). The median height was 166.5 cm (IQR 12 cm), and the median weight was 72 kg (IQR 11 kg), for a median BMI of 26.23 kg/m2 (IQR 3.64 kg/m2). Participants were taking a median of 3 current psychotropic medications (IQR 2 medications) and had a median duration of illness of 7 years (IQR 4 years). The estimated energy requirement was 2055 kcal for females (IQR 138 kcal) and 2678 kcal for males (IQR 377 kcal). Of the 150 participants, 69 (46%) were male, and 129 (86%) were single. A family history of mental illness was reported in 76 (51%) participants. The most common psychiatric disorders were bipolar affective disorder in 50 (33.3%) participants; major depressive disorder in 50 (33.3%) participants; and schizophrenia and related disorders in 50 (33.3%) participants. See Table 1”.
However, we request to retain Table 1 to make information visible for the readers and also to future meta-analysts.
- Please state the duration of questionnaire’s response.
Authors’ response: We added the following: “The questionnaire required approximately 15-20 minutes to complete”.
Discussion:
- The two first sentences are redundant.
Authors’ response: We deleted the sentence: “The FFQ appears to be an adequate tool for ranking nutrient intakes in this population but may overestimate absolute intakes” to remove redundancy.

Reviewer 2 Report
Comments and Suggestions for Authors
Dear Author,
your efforts for writing manuscript are impressive but following comments need to do
· 1. Initial part of Abstract is week “Patients with serious mental illnesses (SMI) like schizophrenia, bipolar disorder, and ma- 28 jor depression have high rates of nutritional deficiencies and obesity” it need to change as per contect you want to deliver in manuscript.
· 2. DIETQ-SMI need to explain in a scientific way with effective literature support.
· 3. This statement is not correct” Patients with serious mental illness (SMI), such as major depressive disorder, schiz- 47 ophrenia and bipolar disorder, are at high risk for nutritional deficiencies and obesity” need to rewrite and corelate with reference.
· 4. In material and methods “Traditional Bahraini cuisine features dishes like machboos (a spiced rice dish), gursan 109 (a traditional bread), and harees (a wheat and meat porridge)” this is not clear rewrite.
· 5. “The complete list of items is shown in Supplement Material 1” at what basis it is designed.
· 6. In line 141: “The portion sizes 141 were identified based on local plate or bowl size.” standardization of food and culinary is done or not.
· 7. In line 155 to 157 : rewrite the content “This cross-sectional study included a randomly selected sample of 150 adults with 155 mental illness aged 18-64 years and living in the community in Bahrain. Study participants 156 were recruited through the Rehabilitation Services Department of the Psychiatric Hospi- 157 tal, Bahrain”
· 8. In section 2.5. Procedures for validation should give with reference
· 9. In line 206 Kindly show the ethics letter or certificate
· 10. Rewrite the results with graphs and chart
· 11. Discussion need to explain with effective way of justification with results.
Comments on the Quality of English LanguageNA
Author Response
Dear Author,
your efforts for writing manuscript are impressive but following comments need to do
Authors’ response: Thank you for your thoughtful review of our manuscript and for providing constructive feedback. We appreciate you taking the time to thoroughly evaluate our work and provide suggestions to improve it. We have carefully considered all of the comments you raised during the peer review process. In response, we have made revisions and highlighted them in yellow for ease of re-review.
English language was further polished by the last author Prof. Mary Seeman an internationally recognized psychiatrists who is native speaker.
- 1. Initial part of Abstract is week “Patients with serious mental illnesses (SMI) like schizophrenia, bipolar disorder, and ma- 28 jor depression have high rates of nutritional deficiencies and obesity” it need to change as per contect you want to deliver in manuscript.
Authors’ response: We agreed with the reviewer and we improved the introduction section of the abstract to: “General food frequency questionnaire (FFQs) have not been tailored or validated for individuals with psychiatric disorders. Given unique dietary behaviors in patients with serious mental ill-nesses (SMI), tailored tools are needed. Few FFQs exist for singular disorders like depression/schizophrenia and lacked diverse validation. Thus, we developed and validated a FFQ tailored for individuals with SMI. 150 adults with SMI (schizophrenia, bipolar disorder, or major depression) were recruited in Bahrain”.
- 2. DIETQ-SMI need to explain in a scientific way with effective literature support.
Authors’ response: We provided the following paragraphs:
“While general food FFQs have been widely used in nutrition research, these standard instruments were not designed specifically for individuals with psychiatric disorders [22]. Given the unique dietary behaviors and needs of those with SMI, tailored and validated tools are required to accurately capture the eating habits of this population [17-19]. To date, only a few FFQs have been created for those with SMI [14, 16]. However, these instruments focused solely on singular disorders like depression or schizophrenia. They have not taken a broader view to assess dietary intake across the spectrum of serious mental illnesses. Furthermore, existing questionnaires have been limited by small sample sizes and lacked psychometric validation using data from diverse demographic groups [23, 24].
Existing FFQs have not been tailored or validated, however, for patients with SMI [23, 24]. This is important because the unique challenges faced by individuals with schizophrenia, bipolar affective disorder, and related severe disorders are known to impact their dietary behaviors and nutritional status in unique ways [18, 25, 26]. Psychiatric symptoms, medication side effects, cognitive deficits, substance abuse including caffeine and nicotine (tobacco smoking), residential neighborhood and social isolation can all influence food choices and nutrient intake [14, 27, 28].
Therefore, we considered that an FFQ designed specifically for assessing the diets of patients with major mental illnesses was needed. It would provide clinicians and re-searchers with an important tool for evaluating nutritional status, identifying potential nutrient deficiencies or excesses, and guiding targeted nutritional interventions. The aim of this study was to develop and validate an FFQ for practical dietary assessment in individuals with SMI”.
- 3. This statement is not correct” Patients with serious mental illness (SMI), such as major depressive disorder, schiz- 47 ophrenia and bipolar disorder, are at high risk for nutritional deficiencies and obesity” need to rewrite and corelate with reference.
Authors’ response: We provided more explanations as follow: “While the concept of serious mental illness (SMI) is widely used across research, practice, and policy settings, there is no consensus on its precise definition [1]. SMI, also known as severe mental illnesses, and are a group of mental health disorders that significantly impact a person's function. These disorders are typically chronic, meaning they persist over a long period of time, and they often require ongoing treatment and management [2]. SMI includes major depressive disorder, schizophrenia and bipolar disorder [3], which are the focus of the present research. Patients with SMI, are at high risk for nutritional deficiencies and obesity [4-9].
- 4. In material and methods “Traditional Bahraini cuisine features dishes like machboos (a spiced rice dish), gursan 109 (a traditional bread), and harees (a wheat and meat porridge)” this is not clear rewrite.
Authors’ response: This sentence was confusing because of the Arabic names of dishes. The sentence was removed with no impact on clarity.
- 5. “The complete list of items is shown in Supplement Material 1” at what basis it is designed.
Authors’ response: We provided a sentence to explain that: “The complete list of food and beverage items provided in Supplement Material 1 was developed through an exhaustive review of existing dietary literature and consultation with experts”.
The paragraph now reads as follow: “This cross-sectional questionnaire validation study was conducted at the Psychiatric Hospital, Bahrain. The research protocol adhered to international guidelines for questionnaire design and followed a systematic multi-step development process. A comprehensive list of foods and beverages commonly consumed by patients with SMI was created by reviewing international/national dietary surveys, systematic reviews and me-ta-analyses, and consulting nutrition experts [16, 23, 29-34]. This initial food list contained more than 350 variants of individual food and beverage items. The list was organized into eight categories: grains (GR), fruits (FR), vegetables (VEG), dairy (DRY), meat and proteins (MP), oils and fats (OFI), beverages (BEV), snacks and sweets (SSS), and condiments (COND). The complete list of items is shown in Supplement Material 1. The complete list of food and beverage items provided in Supplement Material 1 was developed through an exhaustive review of existing dietary literature and consultation with experts”.
- 6. In line 141: “The portion sizes 141 were identified based on local plate or bowl size.” standardization of food and culinary is done or not.
Authors’ response: We provided details about standardization as follow: “The portion sizes were identified based on local plate or bowl size. Thus, to further standardize portion estimation, food models and utensils were utilized. Participants were shown plates, bowls, glasses, and other serving ware with examples of small, medium, and large portion sizes as a reference. The use of visual food models aligned with published recommendations for standardizing portion size estimation in food frequency questionnaires [14, 16]. After determining portion size, those who ate small portions were asked to respond with 0.5, and those who ate relatively large portions were asked to respond with 1.5. This approach enabled standardized quantification of portion sizes consumed by participants”.
- 7. In line 155 to 157 : rewrite the content “This cross-sectional study included a randomly selected sample of 150 adults with 155 mental illness aged 18-64 years and living in the community in Bahrain. Study participants 156 were recruited through the Rehabilitation Services Department of the Psychiatric Hospi- 157 tal, Bahrain”
Authors’ response: We rewrite the sentence into: “The study sample consisted of 150 adults aged 18-64 years with mental illness living in the community in Bahrain. Participants were randomly selected from outpatients receiving services through the Rehabilitation Department of the Psychiatric Hospital in Bahrain”.
- 8. In section 2.5. Procedures for validation should give with reference
Authors’ response: We provided details of the procedures for validation as follow: “To evaluate the validity of the FFQ, estimated intakes were compared to 3-day dietary records. Correlation coefficients were used to assess the ability of the FFQ to accurately rank participants' habitual dietary intake [37]. Wilcoxon signed-rank tests were used to compare nutrient intake between the two methods. Pearson’s correlation coefficients were computed to evaluate the correlation between the FFQ score and food diet records [37].
The internal consistency of the FFQ was evaluated using McDonald's omega [38] and Cronbach's alpha [39]. Omega values ≥0.7 indicate good reliability. The test-retest reliability of the FFQ was evaluated by having a subset of participants (n = 50) complete the FFQ a second time, approximately two weeks after initial administration. Test-retest reliability was estimated using intraclass correlation coefficients (ICCs) between nutrient and food group intakes from the two FFQ administrations [40]. ICCs were computed for both un-adjusted intakes. ICC values above 0.40 were considered acceptable, with higher values indicating better reproducibility. The reliability of the FFQ was also evaluated by estimating the proportion of participants classified into the same or adjacent quartile of intake on the two administrations [41]. Excellent reliability was defined as more than 50% of participants classified into the exact same quartile, and substantial reliability was defined as more than 50% classified into the same or adjacent quartile on the repeat administration”.
We added references as:
Schober, P., C. Boer and L. A. Schwarte. "Correlation coefficients: Appropriate use and interpretation." Anesthesia & analgesia 126 (2018): 1763-68.
- Orcan, F. "Comparison of cronbach’s alpha and mcdonald’s omega for ordinal data: Are they different?" International Journal of Assessment Tools in Education 10 (2023): 709-22.
- Tavakol, M. and R. Dennick. "Making sense of cronbach's alpha." International journal of medical education 2 (2011): 53.
- Müller, R. and P. Büttner. "A critical discussion of intraclass correlation coefficients." Statistics in medicine 13 (1994): 2465-76.
- Noor Hafizah, Y., L. C. Ang, F. Yap, W. Nurul Najwa, W. L. Cheah, A. T. Ruzita, F. A. Jumuddin, D. Koh, J. A. C. Lee, C. A. Essau, et al. "Validity and reliability of a food frequency questionnaire (ffq) to assess dietary intake of preschool children." Int J Environ Res Public Health 16 (2019): 10.3390/ijerph16234722.
- 9. In line 206 Kindly show the ethics letter or certificate
Authors’ response: We provide the ethical approval letter as Research Committee of Psychiatric Hospital, Bahrain (PREC/2023/1178) in 17 October 2023. Appended at the end of this Letter.
- 10. Rewrite the results with graphs and chart
Authors’ response: We added an important Figure as follow:
Figure 1. The difference between nutrient estimates from the FFQ versus the FR.
Notes: Positive values indicate the FFQ estimated higher intake than the FR. Negative values indicate the FFQ estimated lower intake. Values near zero signify similar estimates between the two methods.
- 11. Discussion need to explain with effective way of justification with results.
Authors’ response: We improved discussion by explaining that: ““A growing body of research has demonstrated bidirectional relationships between diet quality and mental health outcomes [42]. For example, several studies indicate that diets high in processed foods, saturated fats and refined carbohydrates can negatively impact mood and cognition, whereas diets rich in fruits, vegetables, whole grains and omega-3 fatty acids are associated with reduced risk for depression and other psychiatric disorders [5, 10, 43]. Proposed mechanisms for these effects include modulation of gut microbiota, inflammation, oxidative stress, and neuroplasticity through dietary components that influence immune and metabolic pathways [42]. Further elucidating the di-et-mental health connection in patients with serious mental illness may reveal modifiable risk factors and opportunities for nutritional interventions. This highlights the need for validated tools like our tailored FFQs to accurately capture dietary habits as a first step”.
REC approval

Reviewer 3 Report
Comments and Suggestions for Authors
I would like to begin by congratulating the authors for their study and their publication. I found the article interesting and with a potential for future practice.
I would like to make the following recommendations:
1. SubSection 2.1 would work better in the discussion sections;
2. The inclusion and exclusion criteria would be better written with bullet points or numbers;
3. The reference list should be written using the MDPI criteria;
4. A few short paragraphs regarding the impact of diet on mental health or potential interactive mechanisms could also be useful. Recommended reference: doi: 10.3390/biomedicines11123233
Author Response
I would like to begin by congratulating the authors for their study and their publication. I found the article interesting and with a potential for future practice.
Authors’ response: Thank you for your thoughtful review of our manuscript and for providing constructive feedback. We appreciate you taking the time to thoroughly evaluate our work and provide suggestions to improve it. We have carefully considered all of the comments you raised during the peer review process. In response, we have made revisions and highlighted them in yellow for ease of re-review.
English language was further polished by the last author Prof. Mary Seeman an internationally recognized psychiatrists who is native speaker.
I would like to make the following recommendations:
- SubSection 2.1 would work better in the discussion sections;
Authors’ response: Thank you. We moved contents of subsection 2.1 to the introduction instead of the discussion based on editorial suggestions.
- The inclusion and exclusion criteria would be better written with bullet points or numbers;
Authors’ response: We added numbers for clarity of the inclusion and exclusion criteria.
- The reference list should be written using the MDPI criteria;
Authors’ response: We used EndNote 20 to create and revise all references per MDPI style specific to the journal Brain Sciences. We manually inspected them for clarity and completion.
- A few short paragraphs regarding the impact of diet on mental health or potential interactive mechanisms could also be useful. Recommended reference: doi: 10.3390/biomedicines11123233
Authors’ response: Thank you for this important suggestion and nice reference. We added the following: “A growing body of research has demonstrated bidirectional relationships between diet quality and mental health outcomes [42]. For example, several studies indicate that diets high in processed foods, saturated fats and refined carbohydrates can negatively impact mood and cognition, whereas diets rich in fruits, vegetables, whole grains and omega-3 fatty acids are associated with reduced risk for depression and other psychiatric disorders [5, 10, 43]. Proposed mechanisms for these effects include modulation of gut microbiota, inflammation, oxidative stress, and neuroplasticity through dietary components that influence immune and metabolic pathways [42]. Further elucidating the di-et-mental health connection in patients with serious mental illness may reveal modifiable risk factors and opportunities for nutritional interventions. This highlights the need for validated tools like our tailored FFQs to accurately capture dietary habits as a first step”.

Round 2
Reviewer 2 Report
Comments and Suggestions for Authors
Dear Author,
· The sample size 150 is not efficient to give any conclusion on diet results.
· Introduction need to strengthen in terms of disease information on SEMI
· In line 55 “Patients with a SMI are at high risk for both nutri- 55 tional deficiencies and obesity” need to clarify what type of nutritional deficiencies and obesity.
· In line 71to 73 the paragraph “As a result, individuals with a SMI 71 exhibit 2-3 times greater rates of obesity than does the general population, with subse- 72 quently increased risk of cardiovascular disease, diabetes, dyslipidemia, and other chronic 73 health problems” are not clear. One part author mentioned unemployed then cardiovascular disease and diabetes. All problem can not address with scientific corelation.
· In line no 186 “150 adults aged 186 18-64 years” what is the basis of selection.
· In line no 186 "150 adults aged “cross-sectional study” why this type of study?
· In line 212 Procedures for validation is not scientifically appropriate.
· In line 428 “This study provides important evidence for the validity of a tailored FFQ …” the nutritional deficiency and information are missing in discussion part.
· The offline and mobile based data entry of participants and patients. These methodologies author has covered is not valid for psychiatric patients This may be false or misleading as all three Psychiatric patients not normal.

Need to improve
Author Response
Dear Author,
- The sample size 150 is not efficient to give any conclusion on diet results.
Authors’ response: In our study, we carefully considered these factors and consulted multiple references by expert teams in the field who have conducted similar research. Based on their extensive experience and existing literature, a minimum sample size of 100 was deemed sufficient to achieve meaningful findings and statistical power. These expert teams have successfully conducted studies with similar sample sizes and obtained significant results.
- Cade J., Thompson R., Burley V., Warm D. Development, validation and utilisation of food-frequency questionnaires-a review. Public Health Nutr. 2002;5:567–587. doi: 10.1079/PHN2001318.
- Willett W., Lenart E. Nutritional Epidemiology. Oxford University Press; New York, NY, USA: 1998. Reproduciblity and validity of food-frequency questionnaires; pp. 101–147.
- Introduction need to strengthen in terms of disease information on SEMI
Authors’ response: This was available in the manuscript as follow: “While the concept of serious mental illness (SMI) is widely used across research, practice, and policy settings, there is no consensus on its precise definition [1]. The term SMI refers to severe mental illness and comprises a group of mental health disorders whose effects significantly undermine function. These disorders are chronic, meaning that they persist and require treatment over long periods of time [2]. The term encompasses diagnoses such as major depressive disorder, schizophrenia and bipolar disorder [3], which are the focus of the present research”.
- In line 55 “Patients with a SMI are at high risk for both nutritional deficiencies and obesity” need to clarify what type of nutritional deficiencies and obesity.
Authors’ response: This was available in the manuscript as follow: “Many SMI patients live on restricted food budgets and have limited access to healthy food [5-8]. Causes include unemployment, alienation from family, residence in disadvantaged neighborhoods, and cognitive disability [5-8]. As a result, individuals with a SMI exhibit 2-3 times greater rates of obesity than does the general population, with subsequently increased risk of cardiovascular disease, diabetes, dyslipidemia, and other chronic health problems [14, 15].
Deficiencies in important nutrients, such as β-carotene, folate, vitamin D, and vitamin B12, are common in this population [14, 16]. This is due to poor diet quality, metabolic changes resulting from medications, and inadequate exposure to sunlight [14, 16]”.
- In line 71to 73 the paragraph “As a result, individuals with a SMI 71 exhibit 2-3 times greater rates of obesity than does the general population, with subsequently increased risk of cardiovascular disease, diabetes, dyslipidemia, and other chronic 73 health problems” are not clear. One part author mentioned unemployed then cardiovascular disease and diabetes. All problem can not address with scientific correlation.
Authors’ response: The statement was supported by four top systematic reviews and meta-analyses. The issue of correlation was induced by the reviewer. We only made a scientific statement about well-supported finding.
- Teasdale, S. B., A. S. Mueller-Stierlin, A. Ruusunen, M. Eaton, W. Marx and J. Firth. "Prevalence of food insecurity in people with major depression, bipolar disorder, and schizophrenia and related psychoses: A systematic review and meta-analysis." Critical Reviews in Food Science and Nutrition 63 (2023): 4485-502.
- Teasdale, S. B., S. Moerkl, S. Moetteli and A. Mueller-Stierlin. "The development of a nutrition screening tool for mental health settings prone to obesity and cardiometabolic complications: Study protocol for the nutrimental screener." International Journal of Environmental Research and Public Health 18 (2021): 10.3390/ijerph182111269.
- Teasdale, S., S. Mörkl and A. S. Müller-Stierlin. "Nutritional psychiatry in the treatment of psychotic disorders: Current hypotheses and research challenges." Brain, Behavior, and Immunity - Health 5 (2020): 10.1016/j.bbih.2020.100070. None.
- Teasdale, S. B., K. Samaras, T. Wade, R. Jarman and P. B. Ward. "A review of the nutritional challenges experienced by people living with severe mental illness: A role for dietitians in addressing physical health gaps." Journal of Human Nutrition and Dietetics 30 (2017): 545-53. 10.1111/jhn.12473.
- In line no 186 “150 adults aged 186 18-64 years” what is the basis of selection.
Authors’ response: The term “adults” is 18-64 years range.
- In line no 186 "150 adults aged “cross-sectional study” why this type of study?
Authors’ response: Cross-sectional studies allow researchers to capture a snapshot of dietary patterns and habits at a specific point in time. As FFQs are commonly used to assess long-term dietary intake, and a cross-sectional design provides an opportunity to collect data on a wide range of food items and their frequencies the method was well-justified.
- In line 212 Procedures for validation is not scientifically appropriate.
Authors’ response: We would like to emphasize that in our study, we indeed focused on the two main pillars of validation: reliability (internal consistency) and validity (correlation between the Food Frequency Questionnaire [FFQ] and the reference method). These pillars are widely recognized and accepted within the scientific community as essential components of validation studies for dietary assessment tools. To ensure the reliability of our FFQ, we conducted rigorous internal consistency analyses. This involved assessing the relationships between different items or components of the FFQ to ensure coherence and consistency in measuring dietary intake. By examining the internal consistency, we aimed to ensure that the questions within the FFQ were reliable and provided consistent results. Furthermore, we assessed the validity of our FFQ by comparing it with a reference method, in this case, a Food Record (FR). The correlation between the FFQ and FR is a commonly used method to evaluate the ability of an FFQ to accurately estimate dietary intake. By establishing a strong correlation between the two methods, we demonstrated that our FFQ had the ability to validly assess dietary intake. We provided a robust scientific justification for our validation procedures, drawing upon established methods and practices widely accepted in the field. These validation procedures have been utilized in numerous studies and have contributed to advancing our understanding of dietary assessment methods.
- In line 428 “This study provides important evidence for the validity of a tailored FFQ …” the nutritional deficiency and information are missing in discussion part.
Authors’ response: Our statement did not include any claims about “the nutritional deficiency” brought by the reviewer.
Our objective was to create a dietary FFQ for patients with mental illness and our statement was clear as: “This study provides important evidence for the validity of a tailored FFQ for assessing dietary intake in people with severe mental illness. The DIETQ-SMI demonstrated good internal consistency and acceptable ranking validity against 3-day food records. These findings indicate that the FFQ can be used to appropriately rank the intake of energy, macro- and micronutrients in this specific population”.
- The offline and mobile based data entry of participants and patients. These methodologies author has covered is not valid for psychiatric patients This may be false or misleading as all three Psychiatric patients not normal.
Authors’ response: It is important to emphasize that the term "normal" can be misleading and inappropriate when referring to individuals with psychiatric conditions. Mental health conditions are diverse and affect individuals differently. We acknowledge the unique needs and circumstances of each participant, and we strive to ensure inclusivity and respect throughout our research.
In our study, we made efforts to ensure that data entry methods were accessible and suitable for all participants, including those with psychiatric conditions. We recognize that some individuals may face challenges related to the severity of their illness, cognitive functioning, or other factors. However, it is crucial to emphasize that not all psychiatric patients face the same limitations, and many are fully capable of using mobile devices for tasks such as logging dietary intake.
Comments about English: Extensive editing of English language required.
Authors’ response: The first reviewer mentioned that the paper was “very well written” and the 3rd said the English was fine. The appreciation of language proficiency may be a subjective matter. The reviewer did not provide specific examples to support such a claim.
Reviewer 3 Report
Comments and Suggestions for Authors
I consider that all my comments have been appropriately addresed and that the paper can be published in its current form.
Author Response
Thanks.